# Anomalous boundary correspondence of topological phases

Jian-Hao Zhang[1] and Shang-Qiang Ning[2, *]

[1]*Department of Physics, The Pennsylvania State University, University Park, Pennsylvania 16802, USA*
[2]*Department of Physics, The Chinese University of Hong Kong, Sha Tin, New Territory, Hong Kong, China*

Topological phases protected by crystalline symmetries and internal symmetries are shown to enjoy fascinating one-to-one correspondence in classification. Here we investigate the physics content behind the abstract correspondence in three or higher-dimensional systems. We show correspondence between anomalous boundary states, which provides a new way to explore the quantum anomaly of symmetry from its crystalline equivalent counterpart. We show such correspondence directly in two scenarios, including the anomalous symmetry-enriched topological orders (SET) and critical surface states. (1) First of all, for the surface SET correspondence, we demonstrate it by considering examples involving time-reversal symmetry and mirror symmetry. We show that one 2D topological order can carry the time reversal anomaly as long as it can carry the mirror anomaly and vice versa, by directly establishing the mapping of the time reversal anomaly indicators and mirror anomaly indicators. Besides, we also consider other cases involving continuous symmetry, which leads us to introduce some new anomaly indicators for symmetry from its counterpart. (2) Furthermore, we also build up direct correspondence for (near) critical boundaries. Again taking topological phases protected by time reversal and mirror symmetry as examples, the direct correspondence of their (near) critical boundaries can be built up by coupled chain construction that was first proposed by Senthil and Fisher. The examples of critical boundary correspondence we consider in this paper can be understood in a unified framework that is related to *hierarchy structure* of topological $O(n)$ nonlinear sigma model, that generalizes the Haldane's derivation of $O(3)$ sigma model from spin one-half system.

## I. INTRODUCTION

Symmetry-Protected Topological (SPT) phases, as short-range entangled states of matter, have been the focus of intensive research over the past decades[1]. They represent a new paradigm in the realm of quantum matter, extending beyond Landau's symmetry-breaking framework. A well-known example of an SPT phase in fermionic systems is the topological insulator, which is protected by both time-reversal and charge conservation symmetries[2,3]. Bulk properties of SPT phases have been systematically constructed and classified for interacting bosonic and fermionic systems using various methods[4–17]. Remarkably, one perspective for investigating these intriguing quantum states is the study of anomalous boundary theories associated with SPT phases. This has led to the development of the bulk-boundary correspondence (BBC) theory for SPT phases.

The boundaries of SPT phases can exhibit various intriguing characteristics. That is, they may be gapless, symmetry broken[18], or topologically ordered in three or higher dimensions[19], but they cannot have a unique symmetric gapped ground state. Extensive studies have been conducted on these SPT anomalous boundaries[14,18–32]. A celebrated example is the two-dimensional topological superconductor, which is protected by time reversal symmetry with $\mathcal{T}^2 = -1$. This phase is notable for hosting helical Majorana edge modes that are robust and protected as long as time-reversal symmetry is preserved[33]. Generally, one fundamental question is: What types of boundaries can a SPT phase have? Given a SPT, its boundary can be realized many by different phases. It turns out that even though being different as a first

glance, all such boundaries have the same topological nature to some sense, that is the 't Hooft anomalies associated with the corresponding symmetries[34,35]. To be more precise, an $n$D boundary, which can be either gapless or gapped, can be realized with specific 't Hooft anomalies on the boundary of the corresponding $n + 1$D SPT phase. This motivates the study of 't Hooft anomalies in $n$D quantum theories, including the conformal field theory[20,31,32,36], symmetric gapped phases, such as 2D Symmetry-Enriched Topological Order, etc[23,37–42]. Very recently, the study of 't Hooft anomaly are formulated into a more general framework of generalized symmetry. For example, the 't Hooft anomaly corresponding to 2D SPT is shown to be equivalent to the generalized symmetry, described by the unitary fusion category whose objects and fusion are the group elements and their group multiplication, and the associated 3-cocycle is the F symbol[43,44]. One great significance of 't Hooft anomaly is that it is an invariance under the renormalization group and then provides important non-perturbative knowledge of IR physics even though the corresponding many-body Hamiltonian may be too hard to be attached[45]. For example, the LSM anomaly, one of the 't Hooft anomalies, will forbid the group state from being symmetric uniquely gapped from the only knowledge of each unit cell no matter what Hamiltonian it takes as long as it preserves the symmetry[46–51].

Recently, there has been extensive theoretical[52–77] and experimental[78–81] interest in symmetry-protected topological (SPT) phases protected by crystalline symmetries. Distinguished from internal SPT phases, the boundaries of $d$-dimensional crystalline SPT phases are typically gapped, yet they exhibit protected lower-dimensional

gapless modes at the hinges or corners. This type of topological phase is dubbed higher-order (HO) topological phases[82–91], topological crystalline phases or crystalline SPT. The robustness of the hinge or corner modes against symmetry perturbations reflects the nontrivial topology of the bulk phases, resulting in the bulk-boundary correspondence (BBC) of these topological crystalline phases.

The study of topological crystalline phases has been predominantly explored in free-fermion systems. Due to analytical limitations, a few representative examples of strongly correlated topological crystalline phases have been analytically understood through coupled-wire constructions[90,91]. An established real-space construction of topological crystals has been proposed to construct and classify topological crystalline phases in interacting systems[63–68]. This construction encompasses both bulk and boundary properties, achieved by decorating suitable lower-dimensional SPT or invertible topological phases on the corresponding blocks of the topological crystals, often referred to as "decorated block states" or "decorated topological crystals."

Very interestingly, it was pointed out that there are profound relationships between the crystalline and internal SPT phases. In Ref.[57], the so-called "*crystalline equivalence principle*" (CEP) was conjectured: the classification of crystalline SPT phases with space group $G$ is identical to that of SPT phases protected by internal symmetry in a subtle and abstract way. Therefore, the crystalline SPT (in Euclidean space) and internal SPT protected by the groups that have the symmetry in the same group structure are one-to-one correspondence. The crystalline symmetry $G$ and internal symmetry correspond to each other by relating the spacial orientation reversing symmetry (such as mirror symmetry) to anti-unitary internal symmetry (such as time-reversal symmetry). For fermionic systems, it has been conjectured and justified by enumerations that the crystalline equivalence principle should be applied in a twisted way: spinless (spin-1/2) fermions should be mapped into spin-1/2 (spinless) fermions[64–67,92]. Nevertheless, the crystalline equivalence principle is performed in a rather formal term, and a more physical understanding of this principle is very desired.

In this paper, we focus on the physical properties, especially on the boundary, of crystalline topological phases and SPT phases following the crystalline equivalence principle. Inspired by the bulk correspondence, it is natural to postulate that the Bulk-Boundary Correspondence (BBC) should also adhere to the crystalline equivalence principle. Remarkably, in Ref.[77], the authors demonstrated the existence of such a correspondence relation in 2+1D fermionic topological phases, which they referred to as the "Crystalline-Equivalent Bulk-Boundary Correspondence" (CEBBC). In this work, we extend this correspondence to three and higher-dimensional bosonic systems. For 3D and higher-dimensional SPT phases, it is essential to note that their boundaries may not necessarily exhibit symmetric gapless modes; instead, they can

be symmetrically gapped, which is usually called anomalous Symmetry-Enriched Topological orders (SET). We embark on our journey by exploring the surface SET correspondence between internal SPT phases and their crystalline SPT counterparts. We establish that if a particular topological order can reside on the boundary of an internal SPT phase, it can similarly exist on the boundary of the corresponding crystalline SPT phase, and vice versa. To rigorously establish this surface SET correspondence, we leverage the utility of anomaly indicators, which are fundamental in quantifying the BBC for topological phases. Following this correspondence, we introduce, to the best of our knowledge, new anomaly indicators [such as Eqs. (13) and (14)] for $SO(N) \times Z_2^{\mathcal{M}}$ symmetry indicated by those for the crystalline counterparts $SO(N) \times Z_2^{\mathcal{T}}$ that were obtained by computing the 3+1d partition function on certain manifolds in Ref.[42]. The correspondence between anomalous SET is also utilized in Ref.[93] to construct gapped boundary of 4+1d bosonic SPT beyond group cohomology.

Furthermore, we delve into scenarios where the boundary systems are either in or near a critical state. To address this, we harness the description of the nonlinear sigma model for SPT phases which is a powerful framework for phases of matter in or near criticality. In the pursuit of establishing direct correspondences in these (near) critical surface scenarios, we unveil an intriguing "hierarchy structure" of the $O(n)$ sigma model, which provide a unified understanding of the surface criticality correspondence discussed in this paper. This structure can be traced back to Haldane's derivation of the $O(3)$ nonlinear sigma model, originating from the study of spin-half systems on lattice sites. To the best of our knowledge, such a hierarchy structure of topological nonlinear sigma model are for the first time exposed. In this paper, our illustrative cases primarily revolve around mirror symmetry and time-reversal symmetry, though we also explore rotational systems to offer a comprehensive understanding of the correspondence between crystalline and SPT phases.

The rest of the paper is organized as follows: In Sec. II, we sketch the main idea of the anomalous boundary correspondence of internal SPT and crystalline SPT in Sec. II A and the main result of surface SET correspondence and surface criticality correspondence in Sec. II B and in Sec. II C. In Sec. III, we will mainly discuss the surface SET correspondence in details for time reversal and mirror SPT and also generalize to discuss the cases with additional continuous symmetry in Sec. III C. In Sec. IV and Sec. V, we discuss the surface criticality correspondence of the mirror system and rotation system respectively. We discuss the hierarchy structure of the $O(n)$ sigma model in Sec. IV C that provides a unified understanding of the surface criticality correspondence discussed in this paper. We summarize and outlook in Sec. VI.

## II. OVERVIEW

### A. Generality of anomalous boundary correspondence

It was conjectured that the topological action that describes the (bosonic) crystalline SPT with crystalline symmetry $G_s$ in Euclidean space $\mathsf{R}^4$ is largely classified by[55,57]

$$H^{d+1}(BG_s, U(1)_r) \qquad (1)$$

where $BG_s$ is the classifying space for the group $G_s$ and the subscript $r$ means that for spatial orientation reversing group element, it will act as a complex conjugate on the $U(1)$ coefficient. Recall that the bosonic SPT protected by internal symmetry $G_{int}$ is also largely classified by $H^{d+1}(BG_{int}, U(1)_T)$ where the subscript $T$ means the complex conjugate action on the $U(1)$ coefficient when the group element is anti-unitary[7]. Therefore, it was observed that the classification of the crystalline SPT by $G_s$ is the same as that of internal SPT by $G_{int}$ which is the same group as $G_s$ by identifying the spatial orientation reversing group element in $G_s$ as an anti-unitary element in $G_{int}$. On physical ground, when coupling to an external probe field, either conventional gauge field or spatial gauge field, the topological response is controlled by the same topological class, for example, in $\mathcal{H}^{d+1}(BG, U(1)_\sigma)$ where $\sigma$ conjugate the coefficient if it reverses the space-time orientation. Such an observation can also be generalized to the beyond group cohomology SPT and crystalline SPT. Now this classification correspondence is phrased as the so-called *crystalline equivalence principle*[57]. So in Euclidean space, the crystalline SPT and internal SPT are one-to-one correspondence. This result is also shown by the decorated block state or topological crystal construction for crystalline topological phases[55].

It is well-known that when putting on the open manifold, the internal SPT phase can have a very interesting boundary phase, which carries a symmetry anomaly that can be canceled by the bulk phase[34,35]. Usually, the quantum anomaly of symmetry is called the 't Hooft anomaly, which is defined by obstruction to gauging the symmetry. To carry such an anomaly, the boundary phase can be degenerate or gapless but not symmetric unique gapped. Such a phenomenon is called *boundary-bulk correspondence*.

For crystalline SPT, there is also such a phenomenon, such as the hing mode or corner modes of some crystalline SPT. This will make the nontrivial topology of the bulk phase manifest, leading to the boundary-bulk correspondence for crystalline SPT. We may also call such nontrivial boundary carries quantum anomaly of the symmetry. Even though the obstruction of gauging crystalline symmetry is still rarely discussed in the literature, we can still justify such a statement by adopting a modern definition of the quantum anomaly of symmetry as an obstruction to have a symmetric unique gapped ground state.

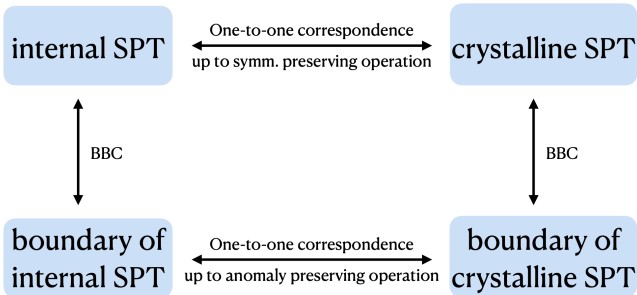

FIG. 1: The anomalous boundary correspondence.

The decorated block state construction is very helpful for understanding the existence of boundary-bulk correspondence of crystalline topological phase in a strongly correlated system. To see this, we first choose one boundary that is called *standard* such that its cell decomposition is one-to-one correspondence to that of the bulk. In fact, such a one-to-one correspondence can be easily established if all the cells in the bulk cell decomposition terminate on the chosen boundary. Then from the decorated block state construction, the decorated nontrivial invertible phase on some lower dimensional cell will terminate on the boundary with nontrivial feature, being gapless or having degenerate ground states. Yet if there might be some low dimensional cells that do not terminate on the chosen boundary, which we call *nonstandard*, then the crystalline topological phase constructed from by decorating these cells would have no nontrivial feature on the boundary due to the decorated invertible phase is not exposed to the boundary. For example, in the two-dimensional $C_N$ rotation system on a disk geometry, there are nontrivial phases by decorating the rotation center with $Z_N$ charge which is at the origin that does not terminate with the boundary that now can be symmetric unique gapped.

Recalling the bulk phases of internal SPT and crystalline SPT are one-to-one correspondence, while the nontrivial boundary states can also manifest the nontrivial topology of the bulk, it is very natural to expect that there is also correspondence between their nontrivial boundary states. In fact, in Ref.[77], the authors studied such correspondence in the two-dimensional fermionic system, which they called "crystalline equivalent boundary bulk correspondence". Here we generalize such a correspondence to higher dimensional and bosonic systems. For example, we will study the surface correspondence between the 3D mirror and time reversal SPT.

The main idea of anomalous boundary correspondence or crystalline equivalent BBC is summarized in Fig.1. We will make a few remarks as below.

1. **Exact anomaly** The quantum anomaly of symmetry is invariant under renormalization flow. So the anomaly specified at the UV limit will always be present in the low energy limit (IR limit). There might be an emergent anomaly of symmetry[94] and

even anomaly of emergent symmetry (in contrast to the exact symmetry that is defined in UV limit), but we do not consider these cases in this paper by assuming the quantum anomaly of exact (not emergent) symmetry is exact, i.e., present in the UV limit. In this paper by symmetry we always mean the exact symmetry.

2. **Symmetry-preserving operation** In the UV limit, the quantum anomaly of symmetry is invariant under symmetry preserving operation (such as tuning parameters of surface Hamiltonian) even though phase transition happens in the ground state[95]. That means two boundary phases with the same quantum anomaly could transfer into each other under by tuning some parameter of the surface Hamiltonian. More generally, one can conjecture that two arbitrary boundary phases with the same quantum anomaly can transform into each other by symmetry allowed operation possibly in an indirect way. One general scheme for such a transformation is through one path of a series of Hamiltonian parameterized by $\alpha \in [0,1]$: $H(\alpha) = \alpha H_0 + (1-\alpha)H_1$ where $H_0$ and $H_1$ realize the two target phases respectively. As both $H_0$ and $H_1$ is symmetric, then the whole path are symmetric, which means the symmetry anomaly in the UV limit is also present. While at the two ends of the path are the two target boundary phases, it is possible along the path some intermediate phase(s) occur.

3. **Topological class of anomaly** For orientation preserving symmetry, one way to see the quantum anomaly of boundary theories is to use the anomaly inflow approach which is cancelled by the presence of the topological bulk[96]. For orientation changing symmetry, there is alternative way to detect the anomaly that is determined by the bulk topology. In other words, the boundary quantum anomaly of symmetry is classified by the topological class of their bulk[34,35]. We say two quantum anomaly of symmetry are equivalent if they correspond to the same topological class, such as $\mathcal{H}^{d+2}(BG_s, Z)$ for bosonic SPT.

4. **Anomaly-preserving operation** The boundary correspondence is assumed to be in the sense of quantum anomaly, namely the boundary correspondence is defined up to anomaly-preserving operation. The anomaly-preserving operation is defined to the transformation that can not change the topological class of the quantum anomaly of symmetry. In fact, the two symmetry anomalies in the boundary correspondence here we discuss are equivalent because the two bulks are subject to the crystalline equivalence principle and controlled by the same topological classes. So the boundary correspondence is nothing but the two boundary theories carrying the same topological class of anomaly can transform into each other by anomaly-preserving operation.

5. **One-to-one correspondence** Together with the above mentioned points that any two theories on the SPT can be transformed into each other by symmetry-preserving operation, then the one-to-one correspondence can be made if we can make a connection (up to some proper anomaly-preserving operation) between arbitrary one boundary theory of the internal SPT and arbitrary one boundary of crystalline SPT.

6. **Application** One important application of the boundary correspondence is that we can explore the quantum anomaly of symmetry by studying its crystalline partner. Recalling that for a phase or theory (such the underlyding topological order of SET), whether it can carry quantum anomaly of certain internal symmetry is equivalent to say whether it can be put on the boundary of SPT protected by that internal symmetry. Then by the boundary correspondence, whether it can be put on the boundary of the SPT is equivalent to whether it can be put on the boundary of the crystalline partner of the SPT. On the other hand, we can also study whether a phase/theory be put on the crystalline SPT by studying whether it can be put on the boundary of internal SPT. Another important application is the guidence for exploring the relation between theories, even for theories live on different dimensions, such as the wire/layer construction for high dimension theory from lower dimensional one (See the examples in Sec.IV C).

## B. Surface SET correspondence

To elucidate the concept of boundary correspondence, we begin by examining scenario where the surfaces of three-dimensional topological phases exhibit topological order with anomalous symmetry, specifically focusing on SET with symmetry anomalies.

Surface correspondence implies that a surface's topological order, denoted as $\mathcal{C}$, can exist on the boundary of topological phases protected by internal symmetry as long as it can also reside on the boundary of the corresponding crystalline topological phases while maintaining the crystalline symmetry, and vice versa. In general, for a given topological order, we can enumerate its symmetry enrichment data, including permutations, symmetry fractionalization, and stacking SPT, for both internal and crystalline symmetries. We then investigate whether there exists at least one anomalous symmetry-enriched pattern for both internal and crystalline symmetries that can be applied to the boundaries of internal SPT and the corresponding crystalline SPT.

To delve into this concept further, we focus our analysis on the surface SET of time-reversal SPT and mirror SPT. It becomes evident that the topological order $\mathcal{C}$ can exhibit a time-reversal anomaly if and only if it can manifest a mirror anomaly. This is demonstrated by establishing that when the topological order $\mathcal{C}$ possesses a nontrivial time-reversal anomaly indicator, denoted as $\eta_{\mathcal{T}} = -1$,

it can also exhibit a nontrivial mirror anomaly indicator, $\eta_{\mathcal{M}} = -1$, and vice versa. (For detailed proofs and additional examples involving $U(1)$, $SU(2)$, and $SO(3)$ or $SO(N)$ symmetries, please refer to Sec.III.)

Additionally, we extend the concept of surface correspondence to another intriguing scenario where both surface theories are either at or near criticality.

### C. Surface criticality correspondence

Now, let's delve into the study of correspondence within the context of surface theories that reside in or near critical states. To illustrate this, we will still use time reversal and mirror SPT states as examples. As previously mentioned, the surface-SET correspondence discussed above establishes a clear one-to-one relationship between the surfaces of time reversal and mirror SPT states. This correspondence demonstrates that all surface theories belonging to the same SPT class can be transformed into one another through appropriate symmetric operations. These operations may involve adjustments to the surface Hamiltonian parameters or coupling to new degrees of freedom. However, our curiosity extends beyond this, as we are keenly interested in establishing a direct connection between the (near) critical surfaces of time reversal, mirror SPT states.

To explore the behavior of (near) critical surfaces, we adopt an alternative description of the bulk SPT theory, one based on the nonlinear sigma model[19,97]. This approach is particularly potent when the theory resides near criticality.

In the context of 3D time reversal SPT (within the framework of group cohomology), it is described by the $O(5)$ nonlinear sigma model, featuring a topological theta term with $\Theta = 2\pi$. What makes the topological sigma model especially convenient is that its surface theory corresponds to the $O(5)$ nonlinear sigma model with a level one Wess-Zumino-Witten (WZW) term, which exhibits fascinating critical behavior[19].

In the case of the 3D mirror bulk, it is constructed by decorating the mirror plane with the 2D Levin-Gu state. This state can be effectively characterized using the $O(4)$ nonlinear sigma model, with a topological theta term of $\Theta = 2\pi$[97]. When the mirror plane terminates at the boundary, a protected gapless mode emerges on the termination line of the mirror plane. This mode is aptly described by the $O(4)$ nonlinear sigma model featuring a level-one WZW term. It is worth noting that the $O(4)$ nonlinear sigma model with a level one WZW term is renowned for exhibiting 1+1D $SU(2)_1$ Conformal Field Theory (CFT) behavior at low energies.

To directly illustrate the surface correspondence in this scenario, we must establish a direct connection between the time reversal surface state and the mirror surface state. The time reversal surface state is described as the 2+1D $O(5)$ sigma model with a level one WZW term, while the mirror surface state corresponds to the 1+1D

$O(4)$ sigma model localized on the mirror termination line, with the rest of the surface remaining trivial.

To transition from the time reversal surface state to the mirror surface state, we introduce a mirror-symmetric magnetic field, denoted as $g(x)\vec{n}_5(x)$, where the function $g(x)$ exhibits mirror symmetry, meaning $g(x) = -g(-x)$ together with mirror action on $\mathcal{M} : \vec{n}(x) \rightarrow -\vec{n}(-x)$. This magnetic field effectively creates a domain wall along the mirror termination line. This domain wall formation is achieved by selecting a specific profile for $g(x)$ such that $g(0) = 0$ and $g(x) \gg 1$ for $x \neq 0$. It can be shown that on this domain wall, the 2+1D $O(5)$ sigma model with a level one WZW term reduces to the 1+1D $O(4)$ sigma model featuring a level-one WZW term.

Transitioning from the mirror surface state to the time reversal surface state presents a more complex challenge. One naive approach involves regarding the 1+1D $O(4)$ theory on the mirror termination line as a reduced description of the 2+1D $O(5)$ theory, utilizing the specific $g(x)$ profile discussed earlier. To recover the 2+1D $O(5)$ theory, one might need to carefully adjust the $g(x)$ profile back to $g(x) = 0$. However, it's crucial to emphasize that this approach has its limitations due to the non-time reversal symmetric nature of the magnetic term $g(x)n_5(x)$. Consequently, it may not provide a entirely valid method for transitioning between the mirror and time reversal surface states while maintaining the required symmetry properties. More sophisticated techniques may be necessary to establish a robust and symmetry-preserving correspondence between these states.

A valid approach to bridging the mirror to time reversal surface states involves the use of a coupled chain construction. Specifically, we can construct the 2+1D $O(5)$ model with a level one Wess-Zumino-Witten (WZW) term by coupling an infinite number of 1+1D $O(4)$ models, each equipped with a level one WZW term, preserving the time reversal symmetry. This construction can be traced back to the pioneering work of Senthil and Fisher[98], where they considered coupled 1+1D $O(4)$ chains with a level one WZW term to construct the 2+1D $O(4)$ sigma model featuring a topological term $\Theta = \pi$, all while preserving the crucial time reversal symmetry.

To complete this construction and establish a direct connection between the mirror and time reversal surface states, it is essential to view the 2+1D $O(4)$ topological sigma model as the 2+1D $O(5)$ sigma model with a level-one WZW term, supplemented by an additional large anisotropic term, such as $(n_5)^2$. Importantly, this anisotropic term maintains time reversal symmetry, ensuring the preservation of symmetry throughout the transition.

The coupled chain construction, stemming from Haldane's work, originally generated the 1+1D $O(3)$ sigma model with a topological term $\Theta = \pi$ by coupling an infinite array of 0+1D spin one-half systems. This construction methodology can be extended to diverse $O(n)$ situations, enabling the creation of an $n$D $O(n+1)$ sigma model with a topological term $\Theta = \pi$ through the cou-

pling of an infinite number of $(n-1)$D $O(n+1)$ sigma models, each endowed with a level one WZW term. By introducing a similar supplemented anisotropic term, the resultant $nD$ $O(n+1)$ model with a topological term $\Theta = \pi$ can be regarded as an anisotropic variant of an $nD$ $O(n+2)$ topological sigma model featuring a level one WZW term. This versatile construction framework, which we call it the *hierarchy structure* of $O(n)$ sigma model, can be applied to establish surface criticality correspondences in various scenarios.

## III. SURFACE SET CORRESPONDENCE FOR 3D TOPOLOGICAL PHASES

Here we will show that the surface SET on the surface of SPT protected by time reversal symmetry and mirror symmetry are one-to-one correspondence, namely the same topological order $\mathcal{C}$ can or cannot carry time reversal anomaly and mirror anomaly simultaneously. We also discuss the cases with additional continuous symmetry such as $U(1)$, $SO(N)$ and $SU(2)$.

### A. Bulk description of time reversal and mirror topological phases

The 3D bosonic SPT protected by time reversal is classified by $Z_2 \times Z_2$[19]. The first root is group cohomology type that is characterized by the effective action[99]

$$S = \frac{1}{2} \int_M w_1 \cup w_1 \cup w_1 \cup w_1 \qquad (2)$$

where $w_i \in \mathcal{H}^i(M, Z_2)$ is the $i$th-order Stiefel-Whitney class. The physics of the this phase is to see that on the codimension-2 time reversal defect, which is a line defect, there is Haldane phase protected by time reversal, whose effective Lagrangian is $\frac{1}{2} w_1 \cup w_1$. There is another equivalent description of this time reversal SPT state as discussed in Sec.II C. The second root state is beyond group cohomolgy, whose effective action is[99]

$$S = \frac{1}{2} \int_M w_2 \cup w_2 \qquad (3)$$

The physics of this state is on the time reversal domain wall (surface defect), there is $E_8$ state.

The classification of 3D mirror SPT in bosonic system on Euclidean space is also $Z_2 \times Z_2$, compatible with the crystalline equivalence principle[55]. From the view of decorated topological crystal, the mirror SPT can be constructed by decorating two dimensional SPT protected by $\mathbb{Z}_2$ symmetry and the invertible $E_8$ state at the mirror plane.

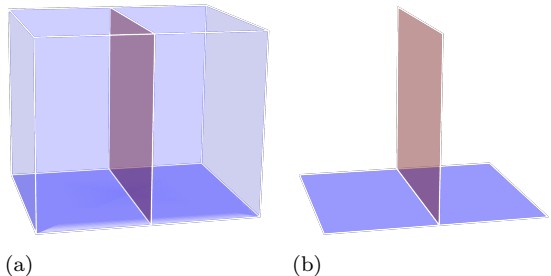

(a)                          (b)

FIG. 2: Mirror plane decoration for 3D mirror SPT. Applying some local unitary transformation, we can also remove the entanglement of left and right wing of the mirror plane (orange color plane) so that it is in trivial product state (as illustrated in (b)). Then the nontrivial mirror SPT is to decorate the mirror plane with (1) $E_8$ state or (2) $Z_2$ internal SPT state. The bottom purple plane terminates the mirror plane is the chosen boundary of the mirror SPT which can be gapless along this mirror line (the line of mirror plane terminating on the boundary) or be (anomalous) SET.

### B. Surface SET and their correspondence

On the surface of 3D SPT phases, it can develop topological order but with anomalous symmetry realization, called anomalous symmetry enriched topological order (SET). Here we will discuss the surface SET correspondence of 3D topological phases protected by time reversal and mirror symmetry. More exmaples see Sec.III C

One convenient way to see the surface SET correspondence is to use the anomaly indicator of the anomalous SET. To characterize the SET with $G$ symmetry anomaly, one need two set of data[100]. The first set is the intrinsic data that is the topological quantities that characterize the topological order when ignoring the symmetry, such as the anyon types $a, b, c...$, quantum dimension $d_a, d_b, d_c...$, topological spin of the anyon $\theta_a, \theta_b, \theta_c$ and so on. The other set is the extrinsic data that characterize the symmetry enrichment of the SET, including how the symmetry permute the anyon, $G$ symmetry quantum number carrying by the anyon, etc. The set of anomaly indicator is a set of expression in terms of these two set of data that can uniquely determine surface of which SPT with the same global symmetry $G$ it can be put on. In other words, the anomaly indicator is just the *quantitative* boundary-bulk correspondence.

The time reversal anomaly indicators of SET on the surface of bosonic time reversal SPT is given by[101]

$$\eta_\mathcal{T} = \frac{1}{D_\mathcal{C}} \sum_{a \in \mathcal{C}} d_a \theta_a \mathcal{T}_a^2. \qquad (4)$$

Here $\mathcal{T}_a^2 = \pm 1, 0$. If the anyon $a$ is invariant under time reversal, i.e., $\rho_\mathcal{T}(a) = a$, $\mathcal{T}_a^2 = \mathcal{T}_{\bar{a}}^2 = \pm 1$ correspond to whether $a$ is locally Kramer degenerate or not. In particular, $\mathcal{T}_1^2 = 1$. If $\rho_\mathcal{T}(a) \neq a$, it is not well-defined to talk

about whether $a$ carries Kramer degeneracy or not, then $\mathcal{T}_a^2 = 0$. When $\rho_{\mathcal{T}}(a) = a$, $\theta_{\mathcal{T}(a)} = -\theta_a$, then $\theta_a = 0, \pi$. Further, if $\rho_{\mathcal{T}}(a) = (\neq)a$, then $\rho_{\mathcal{T}}(\bar{a}) = (\neq)\bar{a}$. When the time reversal anomaly $\eta_{\mathcal{T}}$ takes -1, it is anomalous and can only live on the surface of time reversal SPT.

Meanwhile the mirror anomaly indicator for SET on the surface (See Fig.2) of bosonic topological crystalline phase protected by mirror symmetry is given by[102]

$$\eta_{\mathcal{M}} = \frac{1}{D_{\mathcal{C}}} \sum_{a \in \mathcal{C}} d_a \theta_a \mu_a. \tag{5}$$

Here $\mu_a$ is the mirror eigenvalue related to the anyon $a$. If $\rho_{\mathcal{M}}(a) = \bar{a}$, $\mu_a = \mu_{\bar{a}} = \pm 1$ while $\rho_{\mathcal{M}}(a) \neq \bar{a}$, $\mu_a = \mu_{\bar{a}} = 0$. In particular, $\mu_1 = 1$. When $\rho_{\mathcal{M}}(a) = (\neq)\bar{a}$, $\rho_{\mathcal{M}}(\bar{a}) = (\neq)a$. The indicator $\eta_{\mathcal{M}}$ can only takes values $\pm 1$ and the value $-1$ means it is anomalous and can only live on the surface of mirror SPT.

Now we show that for a topological order $\mathcal{C}$, when $\eta_{\mathcal{T}} = -1$, then we can also have $\eta_{\mathcal{M}} = -1$ for proper mirror symmetry realization. For this, we first divide the anyon of $\mathcal{C}$ into two sets: $A = \{a|\bar{a} = a, a \in \mathcal{C}\}$ and $B = \{a|\bar{a} \neq a, a \in \mathcal{C}\}$. In other words, $A$ consists of anyon that are self-conjugate and $B$ consists of non-self-conjugate one. For example, all the anyons in toric code are self-conjugate but the nontrivial anyons in $\nu = 1/3$ Laughlin state is non-self-conjugate. We can further pick one of each conjugate pair $(a, \bar{a})$ in $B$ to consist of $B_1$. Then the above two anomaly indicators can be expressed into

$$\eta_{\mathcal{T}} = \frac{1}{D_{\mathcal{C}}} \left( \sum_{a \in A} d_a \theta_a \mathcal{T}_a^2 + \sum_{a \in B_1} 2 d_a \theta_a \mathcal{T}_a^2 \right) \tag{6}$$

$$\eta_{\mathcal{M}} = \frac{1}{D_{\mathcal{C}}} \left( \sum_{a \in A} d_a \theta_a \mu_a + \sum_{a \in B_1} 2 d_a \theta_a \mu_a \right) \tag{7}$$

where we have used $d_a \theta_a = d_{\bar{a}} \theta_{\bar{a}}$ and $\mu_a = \mu_{\bar{a}}$ and $\mathcal{T}_a^2 = \mathcal{T}_{\bar{a}}^2$. Now we can first identify $\mathcal{T}_a^2 = \mu_a$ for $a \in A$. If we can further identify $\mathcal{T}_a^2 = \mu_a$ for $a \in B_1$, we can immediately have $\eta_{\mathcal{T}} = \eta_{\mathcal{M}}$. Recall that $\mathcal{T}_a^2$ takes nonzero value only if $\rho_{\mathcal{T}}(a) = a$ while $\mu_a$ nonzero only when $\rho_{\mathcal{M}}(a) = \bar{a}$. Then for such identification to satisfy, it is sufficient if we have for any $\rho_{\mathcal{T}}(a) = a$, we can have a permutation $\rho_{\mathcal{M}}$ for mirror symmetry such that $\rho_{\mathcal{M}}(a) = \bar{a}$. In fact, this can by done if we define $\rho_{\mathcal{M}} = \rho_{\mathcal{K}} \rho_{\mathcal{T}}$ where $\rho_{\mathcal{K}}$ is the charge conjugate operation of a topological order that maps any anyon $a$ into its conjugate $\bar{a}$ while keeping all the topological quantities invariant. Therefore, if we have $\eta_{\mathcal{T}} = -1$ for a topological order, we can also have proper mirror symmetry realization such that $\eta_{\mathcal{M}} = -1$ for the same topological order and vice versa. In other words, a topological order can carry the time reversal anomaly if and only if it can carry the mirror anomaly.

Besides the above time reversal anomaly and mirror anomaly which correspond the group cohomology class, there is still the beyond group cohomology class. For this case, the surface anomalous SET correspondence is

more obvious by checking the anomaly indicator for these states[103]

$$\eta_1 = \frac{1}{D_{\mathcal{C}}} \sum_{a \in \mathcal{C}} d_a^2 \theta_a. \tag{8}$$

This anomaly indicator is for both time reversal and mirror SPT that are beyond group cohomology. As the time reversal and mirror properties do not explicitly enter in the expression of this indicator.

### C. Other examples

#### 1. Topological insulator and topological crystalline insulator protected by mirror symmetry

For 3D bosonic topological insulators with $U(1) \times Z_2^{\mathcal{T}}$ and their crystalline counterparts—topological crystalline insulator with $U(1) \times Z_2^{\mathcal{M}}$, their classification are both $(Z_2)^4$. Two of the roots are protected by only time reversal or mirror symmetry, which can be detected by the above mentioned two anomaly indicators $\eta_1$ and $\eta_{\mathcal{T}}$ or $\eta_{\mathcal{M}}$. The other two roots need the protection jointly by $U(1)$ and time reversal or mirror. For $U(1) \times Z_2^{\mathcal{T}}$, these two roots can be detected by the following two anomaly indicators[104]

$$\eta_{3,\mathcal{T}} = \frac{1}{D_{\mathcal{C}}} \sum_{a \in \mathcal{C}} d_a^2 \theta_a e^{i2\pi q_{\mathcal{T}}(a)} \tag{9}$$

$$\eta_{4,\mathcal{T}} = \frac{1}{D_{\mathcal{C}}} \sum_{a \in \mathcal{C}} d_a \theta_a e^{i2\pi q_{\mathcal{T}}(a)} \mathcal{T}_a^2 \tag{10}$$

where $q_{\mathcal{T}}(a)$ is the fractional $U(1)$ charge carried by anyon $a$, that satisfies $q_{\mathcal{T}}(a) + q_{\mathcal{T}}(b) = q_{\mathcal{T}}(c)$ mod 1 if $N_{ab}^c$ nonzero. We put the subscript $\mathcal{T}$ here to imply this is defined under the present of time reversal symmetry. In particular, $q_{\mathcal{T}}(a) + q_{\mathcal{T}}(\bar{a}) = 0$ mod 1 since $N_{a\bar{a}}^1 = 1$. Even though the time reversal does not explicitly enter in the expression of $\eta_3$, time reversal indeed has effect on $\eta_3$ by enforcing further constraint on the fractional charge $q_{\mathcal{T}}(a)$. As the group structure $U(1) \times Z_2^{\mathcal{T}}$ indicates $\hat{Q}\mathcal{T} = -\mathcal{T}\hat{Q}$, we have $q_{\mathcal{T}}(\rho_{\mathcal{T}}(a)) = -q_{\mathcal{T}}(a)$. So for $\rho_{\mathcal{T}}(a) = a$, we have $q_{\mathcal{T}}(a) = q_{\mathcal{T}}(\bar{a}) = 0, \frac{1}{2}$ mod 1.

For $U(1) \times Z_2^{\mathcal{M}}$, the corresponding two indicators are[105]

$$\eta_{3,\mathcal{M}} = \frac{1}{D_{\mathcal{C}}} \sum_{a \in \mathcal{C}} d_a^2 \theta_a e^{i2\pi q_{\mathcal{M}}(a)} \tag{11}$$

$$\eta_{4,\mathcal{M}} = \frac{1}{D_{\mathcal{C}}} \sum_{a \in \mathcal{C}} d_a \theta_a e^{i2\pi q_{\mathcal{M}}(a)} \mu_a \tag{12}$$

where $q_{\mathcal{M}}(a)$ also satisfies $q_{\mathcal{M}}(a) + q_{\mathcal{M}}(b) = q_{\mathcal{M}}(c)$ mod 1 if $N_{ab}^c$ nonzero and then $q_{\mathcal{M}}(a) + q_{\mathcal{M}}(\bar{a}) = 0$ mod 1. Similarly, the mirror symmetry enforces additional constraint on the $q_{\mathcal{M}}(a)$ that is $q_{\mathcal{M}}(\rho_{\mathcal{M}}(a)) = q_{\mathcal{M}}(a)$. Therefore, for $\rho_{\mathcal{M}}(a) = \bar{a}$, we have $q_{\mathcal{M}}(a) = q_{\mathcal{M}}(\bar{a}) = 0, \frac{1}{2}$.

Now we can see that we can map $\eta_{3,\mathcal{T}}$ and $\eta_{4,\mathcal{T}}$ to $\eta_{3,\mathcal{M}}$ and $\eta_{4,\mathcal{M}}$ respectively by identification $\rho_{\mathcal{M}} = \rho_{\mathcal{K}} \rho_{\mathcal{T}}$,

$\mu_a = \mathcal{T}_a^2$ as discussed in Sec. III B and also $q_{\mathcal{M}}(a) = q_{\mathcal{T}}(a)$ for all $a \in \mathcal{C}$. The last identification can be checked by $q_{\mathcal{M}}(\rho_{\mathcal{M}}(a)) = q_{\mathcal{T}}(\overline{\rho_{\mathcal{T}}(a)}) = -q_{\mathcal{T}}(\rho_{\mathcal{T}}(a)) = q_{\mathcal{T}}(a) = q_{\mathcal{M}}(a)$ mod 1. The first equality uses the identification $\rho_{\mathcal{M}} = \rho_{\mathcal{K}}\rho_{\mathcal{T}}$. The second equality use $q_{\mathcal{T}}(a) + q_{\mathcal{T}}(\bar{a}) = 0$ mod 1. The third equality uses $q_{\mathcal{T}}(\rho_{\mathcal{T}}(a)) = -q_{\mathcal{T}}(a)$.

Therefore, we can conclude that if a topological order with time-reversal symmetry can be put on the boundary of the 3D topological phases protected by $U(1) \times Z_2^{\mathcal{T}}$, it can also be put on the boundary of the 3D topological phases protected by $U(1) \times Z_2^{\mathcal{M}}$ and vice versa. This just builds up the surface anomalous SET correspondence to a bosonic topological insulator by $U(1) \times Z_2^{\mathcal{T}}$ and topological crystalline insulator by $U(1) \times Z_2^{\mathcal{M}}$. The results for $U(1) \rtimes Z_2^{\mathcal{T}}$ and $U(1) \rtimes Z_2^{\mathcal{M}}$ can be discussed similarly.

### 2. Topological phase with $SU(2) \times Z_2^{\mathcal{T}}$ and topological crystalline phase with $SU(2) \times Z_2^{\mathcal{M}}$

The classification for 3D bosonic topological phase protected by $SU(2) \times Z_2^{\mathcal{T}}$ and $SU(2) \times Z_2^{\mathcal{M}}$ is $(Z_2)^3$. The first two roots do not need the protection of $SU(2)$ and have already been discussed above. So we focus on the third root here. As the phase is still protected by breaking the $SU(2)$ down to $U(1)_z = \{e^{i\alpha \hat{S}_z} | \alpha \in [0, 4\pi)\}$ where $\hat{S}_z = \frac{\hat{\sigma}_z}{2}$. So we can utilize the above set of indicators for $U(1)$ case. As it was shown in Ref.[105] we can use $\tilde{\eta}_{3,\mathcal{M}} = \eta_1 \eta_{3,\mathcal{M}}$ to detect to third root state, which always takes the trivial value $\tilde{\eta}_3 = 1$ for all possible topological order. That leads to the result that there is the so-called "symmetry enforced gaplessness" phenomenon on the surface of the topological phase protected jointly by $SU(2) \times Z_2^{\mathcal{M}}$. On the other hand, it is also shown that there is also "symmetry enforced gaplessness" on the surface of topological phase protected jointly by $SU(2) \times Z_2^{\mathcal{T}}$. This is consistent with the surface SET correspondence: no SET can carry this anomaly.

### 3. Topological phase with $SO(N) \times Z_2^{\mathcal{T}}$ and topological crystalline phase with $SO(N) \times Z_2^{\mathcal{M}}$

The classification for 3D topological phases protected by $SO(3) \times Z_2^{\mathcal{T}(\mathcal{M})}$ is classified by $(Z_2)^4$. Two roots of them are protected only by time reversal. And the other two ones are one-to-one correspondence to the ones by breaking $SO(3)$ down to $U(1)_z = \{e^{i\alpha \hat{S}_z} | \alpha \in [0, 2\pi)]\}$ where $\hat{S}_z$ is the integer spin $z$-component operator. It was shown in Ref.[105] by replacing $q(a) \to s_a^z$, we have two anomaly indicators for these two root phases for $SO(3) \times Z_2^{\mathcal{M}}$

$$\eta_{3,\mathcal{M}} = \frac{1}{D_{\mathcal{C}}} \sum_{a \in \mathcal{C}} d_a^2 \theta_a e^{i2\pi s_a} \tag{13}$$

$$\eta_{4,\mathcal{M}} = \frac{1}{D_{\mathcal{C}}} \sum_{a \in \mathcal{C}} d_a \theta_a e^{i2\pi s_a} \mu_a \tag{14}$$

where we used $s_a^z = s_a$ mod 1 and $s_a$ is the $SO(3)$ spin carried by anyon $a$.

Following similar reasoning above, we can conjecture the two corresponding anomaly indicators for 3D topological phases protected by $SO(3) \times Z_2^{\mathcal{T}}$ are

$$\eta_{3,\mathcal{T}} = \frac{1}{D_{\mathcal{C}}} \sum_{a \in \mathcal{C}} d_a^2 \theta_a e^{i2\pi s_a} \tag{15}$$

$$\eta_{4,\mathcal{T}} = \frac{1}{D_{\mathcal{C}}} \sum_{a \in \mathcal{C}} d_a \theta_a e^{i2\pi s_a} \mathcal{T}_a^2. \tag{16}$$

Interestingly, these were proved using a different approach in Ref.[42] where these two formulas are also shown to be the same for general $SO(N) \times Z_2^{\mathcal{T}}$[106]. So in turn, we can conjecture the two anomaly indicators (13) and (14) are also the same for $SO(N) \times Z_2^{\mathcal{M}}$ which might be derived using the folding trick in Ref.[105].

Therefore, the topological order $\mathcal{C}$ can or cannot be put on the surface of 3D topological phases protected by $SO(N) \times Z_2^{\mathcal{T}}$ and $SO(N) \times Z_2^{\mathcal{M}}$[107]. This builds up the surface SET correspondence of them.

## IV. SURFACE CRITICALITY CORRESPONDENCE OF 3D TOPOLOGICAL PHASES: MIRROR SYSTEM

In this section, we discuss the (near) critical surface of 3D mirror symmetry-protected topological phase and also its counterpartner—3D time-reversal topological phase—that is classified by group cohomology.

### A. The near critical bulk description of mirror and time reversal topological phase

As mentioned in Sec. III A, the 3D bosonic mirror SPT can be constructed by decorated the mirror plane with the Levin-Gu state, the $Z_2$ bosonic SPT. We note that there are a few effective theories to characterize the Levin-Gu state, such as Chern-Simons theory and non-linear sigma model (NL$\sigma$M). Here we will adopt the non-linear sigma model because it is a powerful theory for the phases at or near criticality. In 2D the Levin-Gu state can be characterized by the O(4) NL$\sigma$M with $\Theta$ term[97]

$$\mathcal{L}_{\text{NLSM}} = \frac{1}{g} (\partial_\mu \vec{n})^2 + \mathcal{L}_\Theta \tag{17}$$

where $\mathcal{L}_\Theta$ is the topological $\Theta$ term

$$\mathcal{L}_{\text{2D}}^\Theta = \frac{i2\pi}{\Omega_3} \epsilon_{abcd} n^a \partial_x n^b \partial_y n^c \partial_\tau n^d \tag{18}$$

with $\Omega_3$ the area of unit 3-sphere. The on-site $\mathbb{Z}_2$ symmetry acts on the vector

$$\mathbb{Z}_2: \ \vec{n} \mapsto -\vec{n} \tag{19}$$

The vector field $\vec{n}$ is specified on the interface[108]. As $g \to \infty$, the ground state of the $n$d NL$\sigma$M is disordered and symmetric.

Furthermore, the time reversal SPT classified by group cohomology can also described by the NL$\sigma$M[97]. Recall that the classification of odd spatial dimensional time-reversal SPT by group cohomology is $Z_2$. The $(n+1)$d (odd $n$) system with time-reversal symmetry $\mathbb{Z}_2^T$ which can be described by an $O(n+2)$ NL$\sigma$M with a topological $\Theta$ term with $\Theta = 2\pi$. In 1+1d, it is the well-known $O(3)$ NL$\sigma$M with $\Theta$ term which describes the Haldane phase. In 3+1d, the time-reversal bosonic SPT by group cohomology can be characterized $O(5)$ NL$\sigma$M Eq. (17) with $\Theta$ term

$$\mathcal{L}_{3D}^{\Theta} = \frac{i2\pi}{\Omega_4} \epsilon_{abcde} n^a \partial_x n^b \partial_y n^c \partial_z n^d \partial_\tau n^e \tag{20}$$

where $\Omega_4$ is the unit 4-sphere area. The time-reversal symmetry to the $O(5)$-vector $\vec{n}$ is

$$\mathbb{Z}_2^T: \ \vec{n} \mapsto -\vec{n} \tag{21}$$

### B. Critical surface theory

Now we turn to discuss the (near) critical surface theory of the 3D mirror and time reversal symmetric SPT.

First, we investigate the boundary of the mirror SPT by truncating the block states with an open surface, which is chosen to be perpendicular to the mirror interface. Therefore, the two surface 2 cells are the boundary of the bulk 3 cells and the 1 cell is that of the bulk 2 cell. This surface is to some sense *standard* since its cell decomposition is one-to-one correspondence to the bulk cell decomposition.

Now we are interested in the surface where both the 2-cell are trivially symmetric gapped out but there is a gapless mode on the 1-cell, i.e., the surface mirror interface, that is now described by 1+1d $O(4)$ NL$\sigma$M with the Wess-Zumino-Witten (WZW) term (see Fig.2)

$$\mathcal{L}_{2D}^{WZW}[\vec{n}] = \int du dx \frac{i2\pi}{\Omega_3} \epsilon_{abcd} n^a \partial_u n^b \partial_x n^c \partial_\tau n^d. \tag{22}$$

The WZW term involves an extension of $O(n+1)$ vector $\vec{n}(\boldsymbol{r}, \tau)$ to $\vec{n}(\boldsymbol{r}, \tau, u)$, such that

$$\vec{n}(\boldsymbol{r}, \tau, u = 0) = (0, \cdots, 1), \ \vec{n}(\boldsymbol{r}, \tau, u = 1) = \vec{n}(\boldsymbol{r}, \tau) \tag{23}$$

In other words, the topological boundary theory of the mirror-symmetric SPT phase is the 1+1d $O(4)$ NL$\sigma$M with WZW term on the mirror interface of the boundary of the system. The 1+1d NL$\sigma$M with WZW term (if the $O(4)$ symmetry does not break) describes the so-called

$SU(2)_1$ CFT in 1+1d, so typically there are protected gapless modes along the surface mirror domain wall. We note that such a boundary theory is derived from the viewpoint of a topological crystal where away from the mirror interface it is disentangled. However, in reality, the nontrivial entanglement can spread over the whole system, not just confine at the interface.

Then we turn to the surface of 3D time-reversal invariant SPT phases which is well-known and can be described by the 2+1d $O(5)$ NL$\sigma$M with level one WZW term

$$\mathcal{L}_{3D}^{WZW} = \int du dx dy \frac{i2\pi}{\Omega_4} \epsilon_{abcde} n^a \partial_u n^b \partial_x n^c \partial_y n^d \partial_\tau n^e \tag{24}$$

With preserving time-reversal, we have $\langle n_5 \rangle = 0$ which can be treated as zero so that we can integrate the $n_5$ component and obtain the $O(4)$ NL$\sigma$M with topological $\Theta$ term (18) but now $\Theta = \pi$ in contrast to $2\pi$, with which values symmetric gapped unique ground state is possible. However, in fact, the $\Theta = \pi$ prevents the boundary from being featureless, i.e., having a symmetric gapped unique ground state.

Now we discuss the correspondence between surface criticalities. The surface of time reversal SPT is described by the 2+1d $O(5)$ level-one WZW term with time-reversal symmetry (21) while the surface of mirror SPT is $O(4)$ level-one WZW term (i.e., 1+1d $SU(2)_1$ CFT) with $\mathbb{Z}_2$ symmetry (19) on the mirror interface. The correspondence between these two surfaces means that they can be connected to each other under proper operation that are allowed by symmetry.

The above correspondence implies, on the one hand, we can obtain 1+1d $SU(2)_1$ CFT at the mirror interface from the 2+1d $O(5)$ NL$\sigma$M with WZW term in a mirror symmetric way. In fact, this is easily done by adding a magnetic field $g(x)n_5(x)$ with $g(x) = -g(-x)$ and choosing the profile of $g(x): g(0) = 0$ and $|g(x)| \geq 1$ for $x \neq 0$. On the other hand, it implies that we can construct the 2+1d $O(5)$ NL$\sigma$M with WZW term in a time-reversal invariant manner according to (21) from the $SU(2)_1$ CFT which are located at the mirror interface. Indeed, such construction does exist which was done in Ref.[98]. Roughly speaking, the authors started from the infinite coupled spin-1/2 chain described by the $SU(2)_1$ CFT which in fact has an emergent symmetry $O(4) \cong SU(2)_L \times SU(2)_R$, and turned on the interchain coupling which is also $O(4)$ invariant and followed the way similar to Haldane's well-known derivation for $O(3)$ $\Theta$ term in 1+1d spin chain, they obtained the 2+1d $O(4)$ NL$\sigma$M with $\Theta$ term where $\Theta = \pi$, which is equivalent to the $O(5)$ NL$\sigma$M with WZW term when preserving the time-reversal symmetry. Below we will review such construction in more detail and then generalize to general cases.

### C. Coupled chain approach for $O(5)$ WZW and generalization to $O(n)$ case

To see the Senthil-Fisher's coupled chain construction $O(5)$ WZW[98], we will first review Haldane's derivation for 1+1d O(3) $\Theta$ term from WZW term that can be generalized straightforwardly to $O(n)$ case. Then $n = 5$ gives us the Senthil-Fisher's construction.

#### 1. Review on Haldane's derivation

To begin with, we first review that effective description of 0+1d spin one-half system that is the 0+1d $O(3)$ nonlinear sigma model with level one WZW term

$$S_0 = \int d\tau \frac{1}{g}(\partial_\tau \vec{n})^2 + 2\pi k \Gamma[\vec{n}(u,\tau)] \tag{25}$$

We have level $k = 1$ and $\Gamma[\vec{n}(u,\tau)]$ is the 0+1D WZW

$$\Gamma[\vec{n}] = \frac{1}{4\pi} \int du d\tau \vec{n} \cdot (\partial_u \vec{n} \times \partial_\tau \vec{n}) \tag{26}$$

where we have extended the vector field $\vec{n}$ from $\tau$ dependent to $(u,\tau)$ with $u \in [0,1]$ such that $\vec{n}(0,\tau) = \hat{z}$ and $\vec{n}(1,\tau) = \vec{n}(\tau)$. The extension of $\vec{n}(u,\tau)$ is a choice of the convention as one can use the south pole instead of the north one at the position of $u = 0$. It is easy to see that $\Gamma[-\vec{n}] = 4\pi - \Gamma[\vec{n}] = -\Gamma[\vec{n}]$ mod $4\pi$. We note that the $4\pi$ here is nothing but the volume of unit $S^2$.

Now we consider a chain consisting of an infinite number of spin one half $\vec{n}_i$ and turn on the antiferromagnetic coupling between two nearest neighborhood spin one half. Then the theory is effectively given by

$$S_1 = \int d\tau \sum_i S_0[\vec{n}_i(\tau)] + u \int d\tau \sum_i \vec{n}_i(\tau) \cdot \vec{n}_{i+1}(\tau) \tag{27}$$

with $u > 0$. The antiferromagnetic coupling drives the nearest neighboring spin to polarize in the opposite direction, namely

$$\vec{n}_i = (-1)^i \vec{n}'_i \tag{28}$$

where $\vec{n}'$ are smooth varying fields. Substitute this ansatz into the intersite coupling $u$, we have

$$u \int d\tau \sum_i \vec{n}_i \cdot \vec{n}_{i+1} \simeq \frac{au}{2} \int dx d\tau (\partial_x \vec{n}'(x,\tau))^2 \tag{29}$$

Now we turn to the term $S_0[\vec{n}_i]$. The first term in $S_0$ combined with the inter-site coupling just discussed above, we arrive at

$$S_1^0 = \frac{1}{ag} \int dx d\tau \frac{1}{v_1^2}(\partial_x \vec{n}')^2 + (\partial_\tau \vec{n}')^2 \tag{30}$$

where $v_1 = 1/(a\sqrt{ug})$.

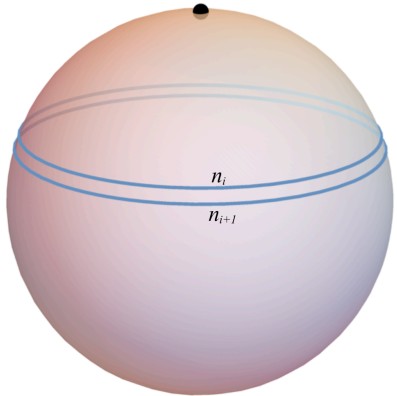

FIG. 3: The n-Sphere angles traced by two configurations of $n_i(\vec{x},\tau)$ and $n_{i+1}(\vec{x},\tau)$ sweeping from $\tau = 0$ to $\tau = \beta$. (Here we use $n_i$ instead of $n'_i$.) The shaded "narrow ribbon" is equal to the difference between the sphere angles, namely $\Delta\Gamma_i$.

Now we turn to the WZW terms in $S_0$. We follow Haldane's approach to derive the topological Theta term from the summed alternative WZW terms. Due to the ansatz Eq.(28) and $\Gamma[-\vec{n}] = -\Gamma[\vec{n}]$ mod $4\pi$, the summed WZW terms becomes

$$\sum_i \Gamma[n'_{2i}] - \Gamma[n'_{2i-1}]$$
$$= \frac{1}{2} \int \frac{dx}{a} \frac{\delta\Gamma[\vec{n}'(x,\tau)]}{\delta\vec{n}'} \cdot \partial_x \vec{n}' a$$
$$= \frac{1}{8\pi} \int dx d\tau \, \vec{n}' \cdot (\partial_\tau \vec{n}' \times \partial_x \vec{n}')$$
$$= \frac{1}{2}\mathcal{I}[\vec{n}] \tag{31}$$

where we have used[109]

$$\delta\Gamma[\vec{n}] = \frac{1}{4\pi} \int d\tau \delta\vec{n} \cdot (\vec{n} \times \partial_\tau \vec{n}). \tag{32}$$

$\mathcal{I}[\vec{n}]$ is the winding number from $S^2$ spanned by $(x,t)$ to $S^2$ spanned by $\vec{n}$. We note that in the continuum limit, $\Gamma[n'_{2i}] - \Gamma[n'_{2i-1}] = \Gamma[n'_{2i-1}] - \Gamma[n'_{2i-2}]$, which lead to the factor $\frac{1}{2}$. Therefore, the second term of summed $S_0[n_i]$ gives the $O(3)$ topological theta term with $\Theta = \pi$.

Now we give a more intuitive derivation of Eq.(31). Recall that the physical meaning of $\Gamma[\vec{n}'_i]$ is the surface angle on the sphere surrounded by the trajectory of physical vector $\vec{n}'_i(\tau)$ from $\tau = 0$ to $\tau = \beta$. Then $\Gamma[\vec{n}'_{2i}] - \Gamma[\vec{n}'_{2i-1}]$ is just the difference of surface angle of the two nearest vectors $\vec{n}'_{2i}$ and $\vec{n}'_{2i-1}$. As the field $\vec{n}'$ varies smoothly in space, then the surface angle difference can be represented by a very "narrow ribbon" on the sphere (See Fig.3). For the closed boundary condition, as $i$ starts from the initial site, passes over the chain, and then back to the initial site, the "narrow ribbon" just smoothly moves on the sphere and then back to the original position. In the continuum limit, it will swap around the

sphere continuously. Therefore, the summation just gives us the winding number of the mapping $\vec{n}' : S^2 \to S^2$. This geometric derivation is convenient to generalize to higher dimensions.

### 2. Generalization to $(n+1)D$

Let us begin with the $nD$ $O(n+2)$ nonlinear sigma model with level one WZW terms

$$S_n = \int dx^n \frac{1}{g}(\partial_\mu \vec{n})^2 + 2\pi k \Gamma_{n+1}[\vec{n}] \qquad (33)$$

where $\vec{n}$ is the $n+2$ component unit vector $k=1$ is the level and $\Gamma_n[\vec{n}]$ is the WZW term

$$\Gamma_{n+1}[\vec{n}] = \frac{1}{\Omega_{n+1}} \int du dx^n \epsilon_{abc...d} n^a \partial_u n^b \partial_x n^c ... \partial_\tau n^d \qquad (34)$$

where $\Omega_{n+1}$ is the volume of $n+1$ sphere. We have extended the vector field $\vec{n}(\vec{x}, \tau)$ to $\vec{n}(u, \vec{x}, \tau)$ such that $\vec{n}(u=0, \vec{x}, \tau) = \hat{x}_{n+2}$ and $\vec{n}(u=1, \vec{x}, \tau) = \vec{n}(\vec{x}, \tau)$. The meaning of $\Gamma_{n+1}[\vec{n}]$ is then the volume (divide $\Omega_{n+1}$) surrounded by the trajectory of $\vec{n}(\vec{x}, \tau)$ on the $n+1$ sphere. Note that the extension of $\vec{n}(\vec{x}, \tau)$ is a convention, as one can also choose the $-\hat{x}_{n+2}$ are the position with $u = 0$. By reversing the direction of $\vec{n}$, the surrounded volume $\Gamma_{n+1}[-\vec{n}]$ is connected to $\Gamma_{n+1}[\vec{n}]$ by $\Gamma_{n+1}[-\vec{n}] = -\Gamma_n[\vec{n}]$ mod $\Omega_{n+1}$.

Now we consider an infinite copy of these $nD$ theories arranged in the $n+1$ (spatial) direction, whose vector fields are denoted as $\vec{n}_i$. Then we turn on the antiferromagnetic coupling between two nearest neighboring vectors $\vec{n}_i$ and $\vec{n}_{i+1}$. So the total action is given by

$$S_{n+1} = \sum_i S_n[\vec{n}_i] + u \int dx^n \sum_i \vec{n}_i \cdot \vec{n}_{i+1}. \qquad (35)$$

with $u > 0$. The antiferromagnetic coupling then drives, similarly to the 1+1D case in Sec. IV C 1, $\vec{n}_i$ and $\vec{n}_{i+1}$ polarize in opposite direction. So we can take the following ansatz

$$\vec{n}_i = (-)^i \vec{n}_i'. \qquad (36)$$

Substitute this ansatz into antiferromagnetic coupling action, very similar to Eq.(29), we have

$$u \int dx^n \sum_i \vec{n}_i \cdot \vec{n}_{i+1} \simeq \frac{au}{2} \int dx^n dz (\partial_z \vec{n}'(x, \tau))^2 \qquad (37)$$

Then the first term in $S_n$ combined with the inter-layer coupling just discussed above, we arrive at

$$S_{n+1}^0 = \int dx^n dz \frac{1}{ag}(\partial_\mu \vec{n}')^2 + \frac{au}{2}(\partial_z \vec{n}')^2. \qquad (38)$$

Now we consider the WZW terms in $S_n$. Noticing the ansatz (36), the summed WZW terms becomes, similarly to 1+1D case,

$$S_{n+1}^1 = \sum_i \Gamma_{n+1}[\vec{n}_{2i}'] - \Gamma_{n+1}[\vec{n}_{2i-1}']$$

$$= \frac{1}{2} \sum_i \Delta\Gamma_{n+1}[\vec{n}_i'] \qquad (39)$$

Each term $\Delta\Gamma_{n+1}[\vec{n}_i']$ is a very "narrow ribbon" on the $S_{n+1}$ sphere (See Fig.3). In the continuum limit, the summation of them gives the winding number $\mathcal{I}[\vec{n}']$ of $S_{n+1}$ by the vector field $\vec{n}'(z, \vec{x}, \tau)$, that is

$$S_{n+1}^1 = \frac{1}{2}\mathcal{I}[\vec{n}']$$

$$= \frac{1}{2\Omega_{n+1}} \int dx^{n+1} \epsilon_{abc...d} n'^a \partial_z n'^b \partial_x n'^c ... \partial_\tau n'^d. \qquad (40)$$

where $\vec{n}' : S_{n+1} \to S_{n+1}$. This is a topological term so it is invariant under any local continuous coordinate transformation, such as the rescaling of one coordinate. Combined Eqs. (38) and (40), and up to proper rescaling of $z$ coordinate, we arrive at

$$S_{n+1} = \frac{1}{ag} \int dx^{n+1}(\partial_\mu \vec{n})^2$$

$$+ \frac{\pi}{\Omega_{n+1}} \int dx^{n+1} \epsilon_{abc...d} n'^a \partial_z n'^b \partial_x n'^c ... \partial_\tau n'^d \qquad (41)$$

which is just the $n+1D$ $O(n+2)$ nonlinear sigma model with theta term $\Theta = \pi$. We note that when $n = 2$, it is just the Senthil-Fisher's construction in Ref.[98] as mentioned in Sec. IV B.

Now we discuss how to obtain the level one WZW term from $\Theta$ term with $\theta = \pi$ while preserving the anomaly. Let us begin with the $1 + 1d$ case. As it is known in 1+1D spin 1/2 chain, there are two equivalent descriptions for the IR physics[110]: the $O(3)$ nonlinear sigma model with $\Theta$ term $\theta = \pi$, and the $O(4)$ nonlinear sigma model with level one WZW term[111]. Even though the direct proof of the equivalence between these two models is not at all straightforward, we utilize another indirect way to argue that they are indeed equivalent (at least in terms of the quantum anomaly of certain symmetry). The way is that we start with the $O(4)$ model, by adding an anisotropic term of one component, says $n^4$, i.e., $-u(n^4)^2$ with ferromagnetic coupling constant $u > 0$[112]. This term preserves the symmetry $n^4 \to -n^4$ and has the effect that the expectation value of $n^4$ is pinned to zero. So we can integrate out the $n^4$ component in $O(4)$ WZW model which exactly gives us the $O(3)$ nonlinear sigma model with $\theta = \pi$. Such a justification of equivalence between the 1+1D $O(3)$ and $O(4)$ model also hints that the anisotropic coupling is irrelevant in the IR limit for the 1+1D case[110].

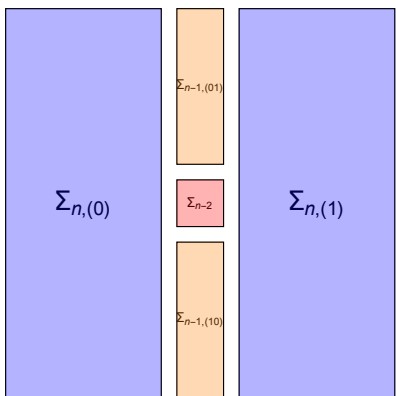

FIG. 4: The cell decomposition of the 2-fold rotation-symmetric systems. The $n$D system with $C_2$ symmetry can be cell decomposed into two $n$ cells $\Sigma_{n,(0)}$ and $\Sigma_{n,(1)}$, two $n-1$ cells $\Sigma_{n-1,(01)}$ and $\Sigma_{n-1,(10)}$ and one $n-2$ cell $\Sigma_{n-2}$. For example, in 2D, there are two 2-cells, two 1-cells, and one 0-cell. Often the (n-2)-cell is also called the rotation center.

For a general $nD$ situation, we can still apply the above anisotropic term argument to relate the $O(n+2)$ $k=1$ WZW model and $O(n+1)$ model with $\theta=\pi$. In terms of focus on the quantum anomaly, as long as the anisotropic term preserves the symmetry, these two theories still have the same anomaly, so such a connection between the two theories is satisfying. We note that while the two theories may maintain the same quantum anomaly, they might not be equivalent in terms of local dynamics in the IR limit. In other words, the possible anisotropic term added may not be irrelevant in the IR limit for general $n>2$ but it seems to be irrelevant for $n=1$.

From the above discussion, we see that it is possible to construct the $nD$ $O(n+1)$ nonlinear sigma model with $\Theta=\pi$ ($nD$ $O(n+2)$ $k=1$ WZW model) that usually maintains quantum anomaly of certain symmetry (such as some symmetry with only order-two element) from lower dimension counterparts, and step by step eventually down to the $0+1D$ case. This messages us that we can track the quantum anomaly down to the $0d$ system. In fact, such a construction/scenario is also shown in other situations, such as the fermion surface anomaly[113] and the LSM anomaly.

## V. MORE EXAMPLES OF SURFACE CRITICALITY CORRESPONDENCE: ROTATION SYSTEM

Consider systems with rotation symmetry $C_N$. We demonstrate $C_2$ symmetry as an example.

## A. Bulk description of rotation systems and their counterpartners

The rotation SPT can be constructed by decorating the rotation center. At the rotation center (see Fig.4), the rotation symmetry acts as a unitary on-site $\mathbb{Z}_2$ symmetry. The rotation symmetric SPT can be constructed by decorating the $(n-2)$D $\mathbb{Z}_2$ SPT at the rotation center. For 0D and 2D, there is one nontrivial $\mathbb{Z}_2$ bosonic SPT (the nontrivial $\mathbb{Z}_2$ charge 0D SPT and 2D Levin-Gu state), while there is none in 1D, which indicates that there is no nontrivial 3D rotation SPT. There is one nontrivial two-fold rotation SPT in 2D and 4D. So here we focus on two cases: 2D and 4D rotation systems. The effective actions of Levin-Gu states are given by Eq. (18) with $\mathbb{Z}_2$ symmetry $\vec{n} \to -\vec{n}$. For 0D $Z_2$ SPT, the effective action $\mathcal{L}_{0D}^\theta$ takes the following form

$$\mathcal{L}_{0D}^\Theta = \frac{i2\pi}{\Omega_1}\epsilon^{ab}n_a\partial_\tau n_b \qquad (42)$$

which together with the $O(2)$ NL$\sigma$M dynamical term $\frac{1}{g}(\partial_\tau\vec{n})^2$ guarantees the ground state is in 0D $\mathbb{Z}_2$ SPT state.

On the other hand, we consider an $(n+1)$d system ($n=2,3,4$) with $\mathbb{Z}_2$ on-site symmetry, which is the crystalline equivalence counterpart of two-fold rotation SPT. The effective field theory is still NL$\sigma$M with topological $\Theta$-term [cf. Eq. (17)], with $\mathbb{Z}_2$ symmetry defined as Eq. (19), which is only well-defined in $(2+1)$d system as a Levin-Gu state[14] and $(4+1)$d system with $O(6)$ vector $\vec{n}$ and topological $\Theta=2\pi$ term

$$\mathcal{L}_{4D}^\Theta = \frac{i2\pi}{\Omega_5}\epsilon_{abcdef}n^a\partial_x n^b\partial_y n^c\partial_z n^d\partial_w n^e\partial_\tau n^f. \qquad (43)$$

where $x,y,z,w$ are the four physical spatial dimensions while $\tau$ is the imaginary time dimension.

The classifications of the $\mathbb{Z}_2$ SPT phases in $(n+1)$D systems are

$$n=2: \ \mathbb{Z}_2; \quad n=3: \ \mathbb{Z}_1; \quad n=4: \ \mathbb{Z}_2 \qquad (44)$$

which is one-to-one corresponding to the classifications of the $C_2$-symmetric SPT phases in $(n+1)$d systems.

We now discuss the (higher) domain defect of SPT protected by internal $Z_2$. For 2D, there is 0D SPT protected by $\mathbb{Z}_2$ at the codimension-2 domain defect. Meanwhile, for 4D, there is 2D SPT protected by $\mathbb{Z}_2$ at the codimension-2 domain defect. This physics can be easily checked using the NL$\sigma$M framework. Interestingly, both decorated rotation center and higher codimension domain defect carry similar nontrivial states.

## B. Critical boundary theories and their correspondence

It is well-known that the topological boundaries of $\mathbb{Z}_2$ SPT are $O(n+2)$ NL$\sigma$M with level-one WZW term [cf.

(22) for 2+1d][97]. The WZW term for $3 + 1$d is given by

$$\mathcal{L}_{3D}^{WZW} = \int_0^1 du \frac{i2\pi}{\Omega_5} \epsilon_{abcdef} n^a \partial_x n^b \partial_y n^c \partial_z n^d \partial_u n^e \partial_\tau n^f. \tag{45}$$

where $O(6)$ vector field $\vec{n}$ are subject to the similar extension as (23).

Now we consider the boundary theory of rotation SPT. The boundary we choose is required to respect the rotation symmetry. For 2D, the systems can naturally be made into a disk $D_2$ whose boundary is just $S_1 = \partial D_2$. However, such a boundary is *non-standard* since it intersects with only the 2-cells and 1-cells but not the rotation center, which means that CD of $S_1$ respect rotation symmetry is not one-to-one correspondence with the bulk CD. For such a boundary, we do not expect there is the surface criticality correspondence. In fact, the boundary theory on $S_1$ can be symmetrically gapped out without any degeneracy. In contrast, the boundary with the same geometry of the 2+1d internal $\mathbb{Z}_2$ SPT can not be gapped out without degeneracy, i.e., is necessary to be gapless or spontaneously symmetry breaking.

For 3D and 4D systems, we can choose a standard boundary, whose structure of CD is one-to-one correspondence to the bulk one. For 3D, we can take the boundary to be perpendicular to the rotation center axis, which is in the same form as Fig. 4, while the bulk extends into the third direction that is not present. The 4D situation is similar.

Physically, with such chosen boundary geometry, the (4+1)d boundary we consider here is trivially symmetrically gapped out almost everywhere but at the boundary rotation center axis there is a gapless mode, i.e., the (1+1)d edge theory that is the (2+1)d Levin-Gu state, described by the $O(4)$ level-one WZW term. As with the standard boundary we do expect there is surface correspondence for 4D $C_2$ rotation systems and internal $Z_2$ SPT.

Now we discuss the correspondence of surface criticality for rotation SPT and $\mathbb{Z}_2$ SPT. We focus on $4 + 1$d. In fact, with the $\mathbb{Z}_2$ symmetry, the six component $n_6$ can be treated as zero and then integrated so that 3+1d $O(6)$ level-one WZW term (45) becomes the 3+1d $O(5)$ topological $\Theta$ term with $\theta = \pi$ in contrast to (20) where $\theta = 2\pi$. The surface correspondence for the $C_2$ rotation system implies that the 1+1d $SU(2)_1$ CFT and $3 + 1$d NL$\sigma$M with topological $\Theta = \pi$ term might have an ultimate connection while preserving symmetry. On the one hand, it is easy to see that the 3+1d NL$\sigma$M with topological $\Theta = \pi$ term reduces to the $O(4)$ NL$\sigma$M with level-one WZW term, which is equivalent to 1+1d $SU(2)_1$ CFT on the codimension-2 $\mathbb{Z}_2$ domain wall.

On the other hand, we should be able to construct the $O(5)$ NL$\sigma$M with $\theta = \pi$ term from the 1+1d $SU(2)_1$ CFT in a $\mathbb{Z}_2$ symmetric manner. We expose the existence of such a construction by recalling the Hierarchy construction of $O(n)$ topological term discussed in Sec.IV C. More explicitly, for the construction, we notice that the

2+1d $O(4)$ NL$\sigma$M with $\Theta = \pi$ term is equivalent to 2+1d $O(5)$ NL$\sigma$M with level-one WZW term if the symmetry $\mathbb{Z}_2 : \vec{n} \to -\vec{n}$ is preserved without changing the surface anomaly. Then the desired construction comes into two steps: First, we use the Senthil-Fisher's construction to obtain 2+1d $O(5)$ NL$\sigma$M with level-1 WZW term for each layer $z \in \mathbb{Z}$ from the coupled $1 + 1$d $SU(2)_1$ CFT; Second, turning on the $O(5)$ invariant coupling which preserves the $\mathbb{Z}_2$ symmetry between different layers, and following the derivation in Sec.IV C, we obtain the 3+1d $O(5)$ NL$\sigma$M with $\Theta = \pi$ which is then equivalent to the 3+1d $O(6)$ NL$\sigma$M with level-one WZW term while preserving the anomaly.

## VI. SUMMARY AND OUTLOOK

In this work, we established the anomalous surface correspondence for internal SPT and crystalline SPT that are subject to the crystalline equivalence principle in three and higher-dimensional bosonic systems, generalizing the so-called *crystalline equivalent bulk-boundary correspondence* for topological phases. We first discuss the correspondence for the surface Symmetry-Enriched Topological order that can appear on the boundary of three or higher-dimensional SPT. For time reversal and mirror SPT, we explicitly show, by utilizing the anomaly indicators, that if one topological order with anomalous time reversal symmetry can live on the boundary of time reversal SPT, it can also live on the boundary of mirror SPT with assigning appropriate mirror symmetry properties and vice versa. We also discuss the SET correspondence including other symmetries which lead us to propose some new anomaly indicators. On the other hand, we also established the direct correspondence between the surface theory in or near criticality by taking the advantage of topological nonlinear sigma model. In pursuit of the direct connection of the (near) critical surface theories, we find the hierarchy structure of the topological $O(n)$ sigma model, originally pioneered by Haldane, that provides a unified way to understand the surface criticality correspondence discussed in this paper. Below we discuss several future problems.

1. Generalize to three and higher fermionic dimension

2. Study the surface criticality correspondence for SPT beyond group cohomology

3. Generalize to SPT phases with average symmetry[114–118].

*Note* − While preparing this manuscript, we notice that the mapping between the time reversal and reflection data in 2D SET are also discussed in Ref.[119].

*Acknowledgements* – Stimulating discussions with Zhen Bi, Ruochen Ma, Liujun Zou, Chong Wang, and Chenjie Wang are acknowledged. JHZ is supported

by the startup fund of the Pennsylvania State University (Zhen Bi). SQN is supported by Direct Grant No.4053578 from The Chinese University of Hong Kong, and funding from Hong Kong's Research Grants Council(GRF No.14306420).

* sqning91@gmail.com

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

[33] Xiao-Liang Qi, Taylor L. Hughes, S. Raghu, and Shou-Cheng Zhang, "Time-reversal-invariant topological superconductors and superfluids in two and three dimensions," Phys. Rev. Lett. **102**, 187001 (2009).

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

[50] Dominic V. Else and Ryan Thorngren, "Topological theory of lieb-schultz-mattis theorems in quantum spin systems," Phys. Rev. B **101**, 224437 (2020).

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

[67] struction and classification of crystalline topological superconductor and insulators in three-dimensional interacting fermion systems," (2022), arXiv:2204.13558 [cond-mat.str-el].

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
