# Peer review of "Anomalous boundary correspondence of topological phases"

_SciPost Physics_

## Round 1 · Referee Report · Anonymous (Referee 1) · 2024-7-2

Strengths

  1. Previously, the bridge between crystalline SPT (protected by crystalline symmetry, e.g., mirror, rotational etc.) and internal SPT (protected by global and internal symmetry) was established in the bulk by the equivalence principle. As shown in Fig.1 of this manuscript, the author steps further to consider the equivalence principle of boundary theories of the two sides. This is an innovative point.
  2. Importantly, the authors study the connection between the two boundary theories by considering the 3D bulk, which raises significant challenges for analytic exploration. The reason is that the 2D boundary of 3D topological phases (SPT here) usually supports diverse candidates of anomalous surface states, e.g., the so-called surface-topological-order that was previously found during the exploration of interacting-topological-insulators of both bosons and fermions. Therefore, the authors study both gapped boundaries and gapless critical surfaces, in order to reach a more complete study.
  3. The authors used various analytic ways to achieve the main results shown, pictorially in Fig.1, including anomaly indicators of both crystalline SPT and internal SPT, NL$\sigma$M, etc., connecting different strategies.

Weaknesses

  1. English writing and grammar should be improved. 2.The work, as indicated by authors in Introduction, relies on the authors' study in Ref.77 which is now a preprint and still in unpublished status. I wonder if it is possible to put these two papers together to make a systematic work for 2D and 3D. This is only a minor suggestion.

Report

This article meets the journal's criteria and I am happy to recommend the publication.

Requested changes

  1. Above eq.2, the citation to Ref. 99 (Teo and Kane's coupled wire construction of non-Abelian quantum Hall states) is possibly a mistake. Please double check. Also above eq.3, the same thing.

  2. On page 3, "1. Exact anomaly". The authors stated that the manuscript avoids the discussions on the potential possibility of anomaly of emergent symmetry. I wonder if the authors can provide more statements on how to avoid this issue, say, in the following analytic exploration (both anomaly indicators and field theory). Is this an in-principle analytically solvable problem or a very hard problem?

Recommendation

Publish (meets expectations and criteria for this Journal)

---

## Round 1 · Referee Report · Anonymous (Referee 2) · 2024-8-4

Strengths

1- The manuscript describes ways to relate symmetry-protected topological (SPT) phases with internal symmetries and SPT with crystalline symmetries. Previous studies make connections between the bulk of such systems. This manuscript describes ways to relate their (anomalous) boundaries. This extends the idea of bulk boundary correspondence to this new physical context. Correspondence between (near) critical boundaries is also discussed.

2- The physical picture of the authors is new to my knowledge.

3- There are various examples that exemplify the correspondence the authors propose.

Weaknesses

1- There are a few places one can (optionally) elaborator further on the generality of the proposed correspondence. This is related to a few clarification questions below.

2- There are a few typos or grammar errors. See suggested minor improvement.

Report

In my view, the manuscript meets the topic selection, and the content meets the standard of SciPost physics. I recommend the manuscript for publication after the authors answer a few questions I have and consider a few suggestions.

Requested changes

Below is a list of clarification suggestions the authors may consider:
1- Can the authors clarify what an “LSM anomaly” is? Presumably, this is related to the Lieb-Schultz-Mattis theorem. However, I cannot extract the definition from the manuscript or the references cited.

2- Above Eq.(6), it is said: “Now we show that for a topological order \mathcal{C}, when $\eta T = −1$, then we can also have $\eta M = −1$ for proper mirror symmetry realization. “

I was not 100 percent sure what the authors meant by “can also have.” Does this follow from an explicit construction, or does it follow from the abstract theory of classification that we can do it? From the context, I would guess the authors meant the latter. It would be nice if this could be made clearer.

3- Below Eq.(7), the authors wrote: “Now we can first identify $T_a^2 =\mu_a$ for $a\in A$. If we can further identify $T_a^2 = \mu_a$ for $ a \in B_1$, we can immediately have $\eta T = \eta M$. “

What does the word "can" mean in this sentence? Again, based on the context, I imagine the authors mean we must be able to find such an identification based on the abstract theory of classification. Is my understanding correct?

4- In the section of III.C titled “Other examples,” each example starts with a sentence (claim) about the bulk classification. Such as “For 3D bosonic topological insulators with $U(1) × Z_2^T$ and their crystalline counterparts—topological crystalline insulator with $U (1) × Z_2^M$ , their classification are both $(Z_2)^4$. ”

As no references were cited, the readers might get the first impression that this is a claim to be explained by the authors’ theory. After a more detailed reading, I believe these results (in the first sentences) are known in previous literature, and the authors use them as the basis to study boundary theory. If this is the case, the authors may consider the clarity of this situation. I will let the authors decide how to improve. I apologize if I misunderstood the purpose of the first sentences (in each example of section III C.)

Below are a few minor grammar errors (typos).

1- Page 1: can be realized many by different phases -> can be realized by many different phases.
2- Left bottom of page 4: defined to the -> defined to be the.
3- Page 7: can by done -> can be done.
4- Bottom of Page 8:“The classification for 3D topological phases protected by $SO(3) × Z_2^{T (M)}$ is classified by $(Z_2 )4$. “ The sentence structure seems not optimal.

Recommendation

Publish (meets expectations and criteria for this Journal)

---

## Editorial Decision

awaiting_resubmission